# RECYCLING SUB-OPTIMIAL HYPERPARAMETER OPTIMIZATION MODELS TO GENERATE EFFICIENT ENSEMBLE DEEP LEARNING

## ABSTRACT

Ensemble Deep Learning improves accuracy over a single model by combining predictions from multiple models. It has established itself to be the core strategy for tackling the most difficult problems, like winning Kaggle challenges. Due to the lack of consensus to design a successful deep learning ensemble, we introduce Hyperband-Dijkstra, a new workflow that automatically explores neural network designs with Hyperband and efficiently combines them with Dijkstra's algorithm. This workflow has the same training cost than standard Hyperband running except sub-optimal solutions are stored and are candidates to be selected in the ensemble selection step (recycling). Next, to predict on new data, the user gives to Dijkstra the maximum number of models wanted in the ensemble to control the tradeoff between accuracy and inference time. Hyperband is a very efficient algorithm allocating exponentially more resources to the most promising configurations. It is also capable to propose diverse models due to its pure-exploration nature, which allows Dijkstra algorithm with a smart combination of diverse models to achieve a strong variance and bias reduction. The exploding number of possible combinations generated by Hyperband increases the probability that Dijkstra finds an accurate combination which fits the dataset and generalizes on new data. The two experimentation on CIFAR100 and on our unbalanced microfossils dataset show that our new workflow generates an ensemble far more accurate than any other ensemble of any ResNet models from ResNet18 to ResNet152.

## 1 INTRODUCTION

Ensemble machine learning is a popular method to use predictions and combine them for a successful and optimal classification.

In the light of its success in Kaggle competition, all top-5 solutions published in the last seven image recognition challenges use at least one ensemble method. The average and median number of individual models used by ensemble is between 7 and 8. Appendix A summarized these 17 solutions.

Despite its recent popularity among practitioners, there is no consensus on how to apply ensemble in the context of deep neural network. The overall work on ensemble Machine Learning (non-deep) was carried out in the 1990s and 2000s. The implementation of Deep Learning on GPU appeared less than 10 years ago. The outbreak of multi-GPU servers allows to effectively train and evaluate many neural networks simultaneously but also deploy ensemble deep architectures.

Another recent trend to improve accuracy is the transfer learning or use external similar data source Kolesnikov et al. (2019). Instead we search a new model-oriented method which can be applied on new kind of problems where no similar dataset exists.

Hyperband-Dijkstra is an innovative way to benefit from this increasing computing power. It consists in unifying the two already proven efficient but contradictory approaches: hyperparameter optimization (HPO) and ensemble. First, one explores and trains models until finding the optimal solution and wasting sub-optimal ones while the other one uses a population of trained models to predict more accurately.

Hyperband-Dijkstra creates an ensemble based on hyperband which is able to generate a huge number of trained deep models. Then, Dijkstra yields efficient combinations between them. As far as we know, it was never proposed to use Dijkstra's algorithm to find a subset of $k$ previously trained models in a greater population.

After that, we describe and discuss interesting properties and experimental results on two datasets:

- Hyperband-Dijkstra is able to generate better ensemble than any ensemble of ResNet models.

- We show that Dijkstra algorithm is better to aggregate $k$ trained models than a naive strategy consisting in taking the top $k$ models based on their validation accuracy.

- We show that our workflow (with ensemble of size $\geq 2$) keeps benefiting of hyperband running after many days while a standard use of hyperband (consisting in taking only the best model) stops improving much earlier.

## 2 RELATED WORKS

In this section we briefly review the main ideas from prior work that are relevant to our method.

**Ensemble.** Authors Sollich & Krogh (1995) laid the foundation stone about the idea that over-fitted machine learning algorithms can be averaged out to get more accurate results. This phenomenon is explained by the Law of Large Numbers which claims that the average of the results obtained from a large number of trials should be close to the expected value. These results are especially interesting for deep learning models because they are machine learning models which are the most affected to random effects (over-fitting) due to their huge amount of parameters.

Many ensemble algorithms have been invented such as Wolpert (1992), Breiman (1996) or boosting Schwenk & Bengio (2000). Some other methods are neural networks specific like negative correlation learning Liu & Yao (1999), dropout Srivastava et al. (2014) or snapshot learning Huang et al. (2017). There is today no consensus on the way to do ensembles like shown in the appendix A.

In case the architecture of models in the ensemble is biased - for example all models contained are not deep enough or not wide enough to capture relevant features in the data - exploiting parametric diversity will not efficiently improve the results. That is why authors Liao & Moody (1999) Gashler et al. (2008) promote more and more diversity, not only based on the random weights initialisation but based on different machine learning algorithms such as neural network and decision tree in the same ensemble to maximize diversity and therefore the accuracy.

**Knapsack problem.** A Combinatorial Optimization problem consists in searching for a solution in a discrete set so that a function is optimized. In many such problems, exhaustive search is not tractable, that is why approximate methods are used. Dijkstra's algorithm Dijkstra (1959) is a path finding algorithm which locally selects the next best node until it reaches the final node. A* Hart et al. (1972) is an informed algorithm which first expands the most promising node to converge faster than Dijkstra. This knowledge is used only if an appropriate heuristic function is available. Otherwise, in absence of this knowledge, Dijkstra and A* are equivalent. More recently, SP-MCTS Schadd et al. (2008) is a probabilistic approach which runs many tree explorations based on the Upper Confident bound applied to Tree (UCT) Kocsis & Szepesvári (2006) formula to guide exploration/exploitation to catch a maximum of information on one node before selecting it.

**Hyperparameter Optimization.** The empirical nature of research in Deep Learning leads us to try many models, optimization settings and pre-processing settings to find the best suited one for data. No Free Lunch theorem Wolpert & Macready (1997) proves that no hyperparameter optimization can show a superior performance in all cases. Nevertheless, methods have been developed and have shown a stable performance on supervised deep learning dataset.

Discrete-space search enables to search the best model description to a given neural network. Under this umbrella, we can find : the number of units per layer, regularization parameters, batch size, type of initialization, optimizer strategy, learning rate. Plenty of approaches exist with a different theoretical background, a pure-exploration approach Bergstra & Bengio (2012), Li et al. (2017), smart computing resources allocation strategies Li et al. (2017) Falkner et al. (2018), *a priori* based Hoffman et al. (2011), *a posteriori* based Bergstra et al. (2011) or genetic inspired Jaderberg et al.

(2017). Those methods are not exclusive, for example BOHB Falkner et al. (2018) mixes Bayesian Optimization strategy to Hyperband.

Another automatic approach exists like graph-space search Pham et al. (2018). It consists in finding the best architecture (graph) of neural networks. It provides a maximum degree of freedom in the construction of the neural network architecture. Due to the infinity of combinations, scientists implement several constraints to limit the possibilities of graph generation, save computation cost and preserve the correctness of generated graphs. All hyper-parameters, like optimization settings and data pre-preprocessing are given by user to drive into this graph-space. Due to this complexity and because only models architectures are explored, we decide to not follow this path.

**Parallel hyperparameter optimization.** All HPO strategies presented in this paper are asynchronous so their deployment is ideal on multi-GPU or multi-node GPU HPC. Distributed client-server softwares Matthew Rocklin (2015) , Moritz et al. (2018) allow to simultaneously spread those training candidate models and evaluate them. Those frameworks allow also serve them in parallel.

**Multi-objective goal.** Authors Johnston et al. (2017) discovered that many neural networks have a comparable accuracy. Literature lets us imagine that the hyper-parameter function topology has two plateaus : where the optimizer algorithm converges and where it does not. This flatness can be used to optimize a secondary goal such as model size, time-to-prediction, power consumption and so on. Authors Patton et al. (2019) propose a multi-objective optimization to not only search an accurate model but also faster ones.

**Early Stopping.** A common practice exists to speed up HPO running like Early Stopping. They consists in resources reallocation strategies by considering learning dynamic of DNN. Prechelt (1998) Li et al. (2017). Early stopping is also known to be a regularization method that stops the training when the validation accuracy plateaus is symptomatic and that it will not generalize well (overfitting).

## 3  PROPOSED WORKFLOW

In this section we will first see the workflow proposed before going into a more detailed explanation step by step.

### 3.1  DETAIL OF THE WORKFLOW

As shown in figure 1, the proposed workflow consists in using hyperband and not only saving the best one on the disk but the sub-optimal one too. Second, a combinatorial optimization algorithm (Dijkstra's) finds the best one regarding the maximum number of models desired by the user (noted $K$). Dijkstra's algorithm computes the validation loss of candidates ensemble to evaluate how well a solution will generalize on the test database.

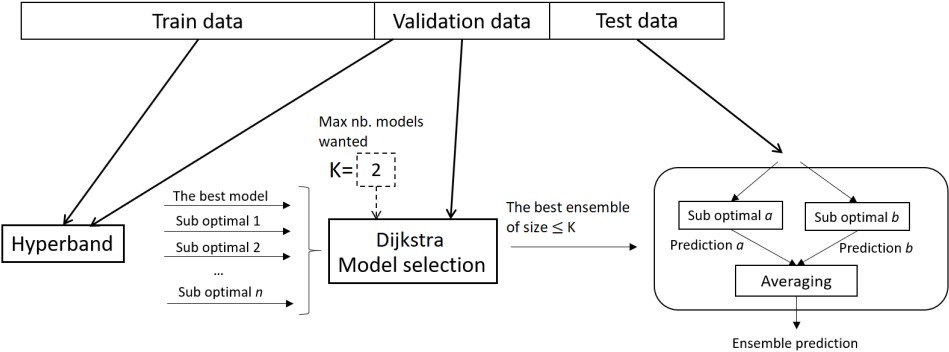

Figure 1: The algorithmic workflow to train, combine and predict on a new data. In this example K=2.

The final accuracy depends on the running time of hyperband and the number of models chosen in the ensemble. Experiments results are shown in section 4.

The workflow we introduce is simple. We use Hyperband algorithm and the distributed framework Ray Moritz et al. (2018) and then our combinatorial optimization Disjkstra's algorithm is a natural choice to ensemble models. The simplicity of the chosen algorithm and re-using existing frameworks reinforce our claims that this work is easy to test on a new dataset.

### 3.2 STEP 1 - HYPERBAND TO GENERATE MANY MODELS

Hyperband relies on an iterative selection of the most promising models to allocate resources, allowing it to exponentially evaluate more configurations than strategies which do not use progressive results during training. Hyperband is a technique that makes minimal assumptions unlike prior configuration evaluation approaches. Its pure-exploration nature combined with conservative resource allocation strategies can sweep better the hyperparameter space than other strategies like blackbox bayesian optimization. This diversity of models sampled are ideal to combine them for Dijkstra's algorithm and make better ensemble Liao & Moody (1999) Gashler et al. (2008).

We only store models trained at least half maximum epochs. This allows to reduce the number of models saved and thus the number of possible combinations by focusing on the most promising models explored by Hyperband.

### 3.3 STEP 2 - DIJKSTRA'S ALGORITHM TO COMBINE MODELS

We discuss that finding the best combination of K among a larger population is first modeled as a graph. We then prove that no exact solution can be applied because of the computing complexity of the problem. That is why we propose Dijsktra's algorithm, a simple and popular approach.

#### 3.3.1 PROBLEM MODELING AND INTUITION

The solution space can be modeled as a tree with the empty ensembles as the root. Every node represents a deep learning model added and any path an ensemble. All nodes can be a terminal and not only leaves. To evaluate and compare their score, we use the formula 1. It calculates the cross entropy between validation labels, averaging predictions and the current ensemble $I$ of size $k$.

$$score_I = CE(y, \frac{1}{k} \sum_{i \in I} \tilde{y_i})$$
(1)

Figure 2 is a simplified example of three models and their combinations on error distribution. The modeled tree associated is shown in figure 3. The problem is to find the best combination to be as close as possible to the center $(0; 0)$. We observe that the best individual model c ($d_{\{c\}} = 0.22$) is not always the best one to combine with the others ($d_{\{a,b\}} = 0.07$). That is why smart algorithms are needed. We also eliminate the simple idea that combining all models systematically leads to the best solution ($d_{\{a,b,c\}} = 0.12$).

#### 3.3.2 PROBLEM COMPLEXITY

Finding the best ensemble of maximum size $K$ among $n$ models with $1 \leq K \leq n$ is a case of the 'knapsack problem' belonging to the NP (non-deterministic polynomial) problem's family. There is no known exact method except brut-forcing all possibilities.

The number of possible subsets of size $k$ among $n$ items is computed with the binomial coefficient. This binomial formula is known Das (2020) to be asymptotically polynomial when $n$ is increased and $k$ is fixed. When the user puts the maximum size of ensembles K to explore, all combinations $k$ such as $1 \leq k \leq K$ are also candidates. Therefore the quantity of candidates ensembles is given by $\sum_{k=1}^{K} \binom{n}{k}$. This formula also has a polynomial behavior when $k$ is fixed and $n$ increases. For example, for $K = 4$ and a population of $n = 100$, adding only one new model in the population increases the number of combinations from 4.09 million to 4.25 million.

This polynomial behavior has two exceptions : when $K = 1$ (linear) and when $K = n$ (exponential). $K = n$ allows the research of big ensembles for a maximum precision despite inference time. The number of ways to construct a non-empty model from a catalogue of N models is formally described by the equation $\sum_{k=1}^{N} \binom{N}{k} = 2^N - 1$. We have two options for each model : using it or not ($2^N$

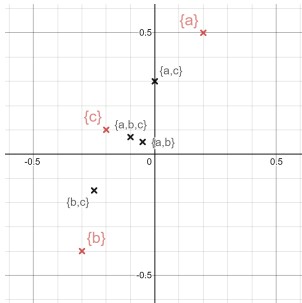

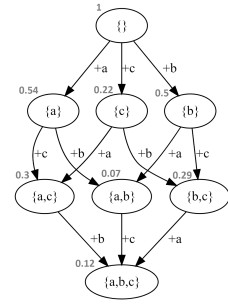

Figure 2: The two axes represent an error of two different classes, the goal is to get closer to the (0;0) point. Each model a, b and c have different error distribution and averaging them leads to ensembles with other error distributions.

Figure 3: Available solutions as a graph. Nodes are ensembles. Edges are the decision to add a new model to the ensemble. The cost function is the euclidean distance and is displayed on the top left of each node. The source node, corresponding to the empty ensemble, is modeled with an arbitrary large distance. The optimal ensemble is made of a and b.

possibilities). We exclude the empty ensemble (-1). It means that for each new model found by hyperband, the quantity of combinations is multiplied by 2. The combination of ensembles with a catalogue of 100 models is $\approx 1.27e^{30}$.

Due to this combinatorial explosion, we understand the need of an approximation search algorithm.

### 3.3.3 APPROXIMATE SOLUTION WITH DIJKSTRA'S ALGORITHM

This huge number of ensembles combined to the fact that relationships between model predictions and labels are complex (figure 2 and formula 1) and that no heuristic is known makes Dijkstra's algorithm a natural choice for this class of problems Voloch (2017). Dijkstra's algorithm is a Dynamic Programming procedure, meaning it makes and memorizes successive approximation choices.

While the number of possibilities requires approximate solutions, this huge number of candidate ensembles has the advantage of ensuring that better combinations should be found compared to a naive aggregation approach. This is confirmed in the results in section 4.

Once a model is found based on the running by Disjkstra's algorithm, we can combine predictions of models on new data or evaluate it on the test dataset. As training and evaluating, models predicting can be distributed on different GPUs but the averaging require all models finished their prediction.

## 4 EXPERIMENTS AND RESULTS

We experiment our workflow on CIFAR100 Krizhevsky (2009) and microfossils datasets both presented in appendix B.3. On these two datasets there are 16 hyper-parameters to explore. Experimental settings are explained in appendices B for reproducibility purpose but this level of detail is not required to understand our works or results. It is possible that larger hyperparameters value ranges may positively influence again the results obtained.

In this section, we evaluate different workflows by evaluating various HPO strategies, different combinatorial optimizations. We also different settings like the number of models in produced ensemble and effect of HPO running time on results.

### 4.1 VISUALIZATION OF ARCHITECTURE SAMPLING RESULTS

As others have highlighted, no correlation exists between the accuracy and computing cost of a model on image recognition. We display it in the figure 4 results of random sampling of hyperparameter space on the CIFAR100 dataset. This is the reason why we propose a target function which measure efficiency of one model based on its accuracy and its inference time. The implemented formula is: $CE(y, \tilde{y}) + WI$ with $I$ the inference time on 2,000 images expressed in seconds and $W$

a scaling factor arbitrary choosen such as $W = 0.001$. Hyperband minimizing this target function increases the concentration of computing resources on more efficient models. The natural efficiency of Hyperband combined to the fact that we use this multi-criteria target function allows to increase the number of models explored in 6 days by factor 3.2 compared to Random Sampling + Early Stopping (plateau detection).

Another Early Stopping method used consist in detecting after one epoch if the models perform better than random predictions on the validation dataset. It shows very effective to detect early which models are unable to learn and free GPUs for other models. Experiments in figure 5 show that about 14% of models diverge so we can save quasi-entirely their running.

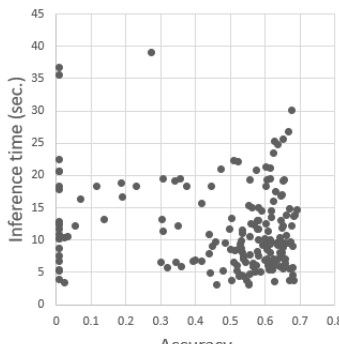

Figure 4: Correlation computing cost versus accuracy of randomly sampled models on CI-FAR100

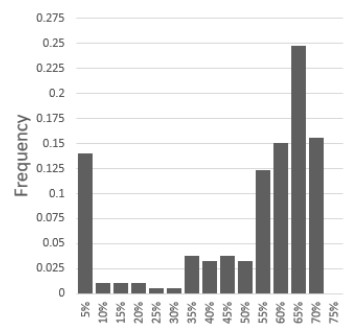

Figure 5: Accuracy histogram of randomly sampled models on CIFAR100

## 4.2 COMPARISON OF VARIOUS HPO STRATEGIES AND DIJKSTRA'S ALGORITHM

We evaluate the accuracy of our workflow on CIFAR100 by replacing Hyperband with various HPO strategies in table 1. Retraining the same deep learning architecture from scratch can yield significant distance in different run time, that is why we compare different HPO strategies and different popular ResNet architectures as well in table 2. We observe that Hyperband generally performs well to take the best one and also to aggregate ensembles compared to all other methods. It confirms our claim in the previous section on Hyperband computing efficiency and the ability to generate good ensembles.

We also observe that most of HPO strategies discovered better models than the best ResNet models found. For both benchmarks, we observe that ResNet18 compared to other ResNet architectures, lead to better models but when we combine them, ResNet34 is a better choice. We conjecture that ResNet18 leads to a lower parametric diversity compared to ResNet34 models because of the lower number of layers.

Another remark is that a proportion of 14% of randomly chosen models diverge while 100% of ResNet models converge. It shows that ResNet are robust handcrafted models but a random process can find more accurate ones on a new dataset.

The same results and conclusion on the microfossils dataset are reached in tables 3 and 4.

Table 1: Various HPO strategies and various ensemble size computed on the CIFAR100 dataset

|  |  | RS | HB | BO | TPE | BOHB | PBT |
|---|---|---|---|---|---|---|---|
|  | Top | 68.74% | **70.69%** | 68.22% | 69.83% | 70.56% | 60.82% |
| Dijkstra solution | Team of 2 | 72.36% | **74.55%** | 71.60% | 73.79% | 74.20% | 65.27% |
|  | Team of 3 | 74.73% | **76.43%** | 73.09% | 74.36% | 76.07% | 67.90% |
|  | Team of 4 | 77.01% | **77.11%** | 73.60% | 75.55% | 77.22% | 68.87% |
|  | Team of 6 | 78.17% | **78.45%** | 75.02% | 76.77% | 77.94% | 69.80% |
|  | Team of 8 | **78.89%** | **78.89%** | 75.56% | 77.33% | 78.16% | 70.73% |
|  | Team of 12 | **79.30%** | 79.18% | 76.04% | 77.25% | 78.60% | 71.09% |
|  | Team of 16 | 79.36% | **79.44%** | 76.04% | 77.34% | 78.70% | 71.39% |

Table 2: Comparison different ResNet populations and ensemble size on the CIFAR100 dataset

|  |  | resnet 18 population | resnet34 population | resnet50 population | resneXt50 population | resnet101 population | resnet152 population |
|---|---|---|---|---|---|---|---|
|  | Top | **65.70%** | 65.23% | 63.37% | 58.72% | 65.15% | 63.18% |
| Dijkstra solution | Team of 2 | 68.02% | **69.30%** | 67.40% | 62.33% | 68.23% | 66.14% |
|  | Team of 3 | 70.12% | **71.42%** | 69.71% | 64.84% | 69.70% | 67.84% |
|  | Team of 4 | 71.34% | **72.37%** | 70.24% | 65.91% | 70.26% | 69.35% |
|  | Team of 6 | 72.30% | **73.12%** | 71.84% | 67.85% | 71.07% | 69.74% |
|  | Team of 8 | 73.02% | **73.79%** | 72.77% | 68.14% | 71.74% | 70.67% |
|  | Team of 12 | 73.41% | **74.22%** | 73.18% | 68.56% | 71.85% | 70.54% |
|  | Team of 16 | 73.44% | **74.26%** | 73.20% | 68.66% | 71.85% | 70.55% |

Table 3: Various HPO strategies and ensemble size on the microfossils dataset

|  |  | RS | HB | BO | TPE | BOHB | PBT |
|---|---|---|---|---|---|---|---|
|  | Top | 86.55% | **88.07%** | 85.56% | 86.09% | 87.82% | 85.10% |
| Dijkstra solution | Team of 2 | 89.02% | **90.18%** | 89.07% | 89.34% | 88.29% | 86.73% |
|  | Team of 3 | 89.16% | **90.68%** | 90.07% | 89.51% | 89.50% | 88.21% |
|  | Team of 4 | 89.77% | **91.09%** | 90.07% | 90.94% | 90.16% | 88.01% |
|  | Team of 6 | 90.30% | **91.13%** | 90.79% | 90.70% | 90.72% | 89.65% |
|  | Team of 8 | 90.61% | **91.54%** | 91.23% | 90.79% | 91.20% | 90.12% |

### 4.3 HYPERBAND AND VARIOUS COMBINATORIAL STRATEGIES

Different combinations algorithms are tested in figures 6 and 7 by varying the number of models from 1 to 16. The total population of models only contains models trained during at least 50 epochs by Hyperband.

We tested two naive strategies. The first one consists in drawing randomly ensembles of $K$ models and the second one in taking the top-$K$. Dijkstra's algorithm generally finds better solutions than naive strategies.

We also evaluate SP-MCTS, a tree search algorithm based on Monte-Carlo. To test SP-MCTS, the solution space was modeled as an unfolded tree representation leading to nodes redundancy, so equivalent nodes were implemented to index the same score. Based on preliminary experiments, SP-MCTS is set to run $1000 \times K$ with $K$ the maximum desired number of models to favor accuracy over SP-MCTS computing cost. With a single-threading implementation, Dijkstra's algorithm takes only 25 seconds to find an ensemble of $K = 16$ among 160 models while SP-MCTS is x580 slower.

In the microfossils dataset, Dijkstra's algorithm falls to a local minimum and uses the same 10 models when $K > 10$. SP-MCTS do not falls into this trap and keeps benefiting of an increasing power.

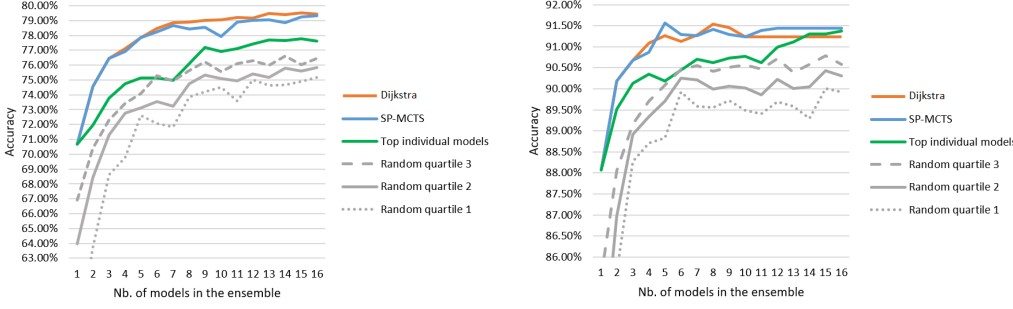

Figure 6: The CIFAR100 dataset

Figure 7: The microfossils dataset

Figure 8: Different combinatorial optimization algorithm tested

Table 4: Various ResNets and ensemble size on the microfossils dataset

| | | resnet 18 population | resnet34 population | resnet50 population | resneXt50 population | resnet101 population | resnet152 population |
|---|---|---|---|---|---|---|---|
| | Top | **86.83%** | 85.24% | 85.65% | 84.91% | 84.22% | 85.09% |
| Dijkstra solution | Team of 2 | 87.74% | **87.77%** | 87.54% | 87.26% | 86.04% | 86.61% |
| | Team of 3 | 88.22% | **88.55%** | 88.48% | 87.93% | 87.54% | 87.66% |
| | Team of 4 | 88.34% | **88.71%** | 88.63% | 88.02% | 87.74% | 87.91% |
| | Team of 6 | 89.10% | **89.50%** | 89.37% | 88.37% | 88.24% | 88.78% |
| | Team of 8 | 89.13% | **89.59%** | 89.39% | 88.73% | 88.40% | 88.80% |

## 4.4 Effect of computing intensity on the final accuracy

Our workflow benefits more of computing intensity than standard Hyperband like shown in figures 9 and 10.

After 24 hours, standard Hyperband (consisting in taking only the best model) converges while our worflow with $K > 2$ keeps benefiting of the models generated. On the CIFAR100 dataset, we identify that ensembles of 12 and 16 models benefit linearly of the computing time. Their accuracy begins to 77% and increases of +0.4% every 24h00.

Moreover, we observe that adding more models systematically leads to an increasing of accuracy but this trend declines. We show that the benefit is obvious from 1 to 2 models (+3.9%) but the improvement is small from 6 to 16 (+1%).

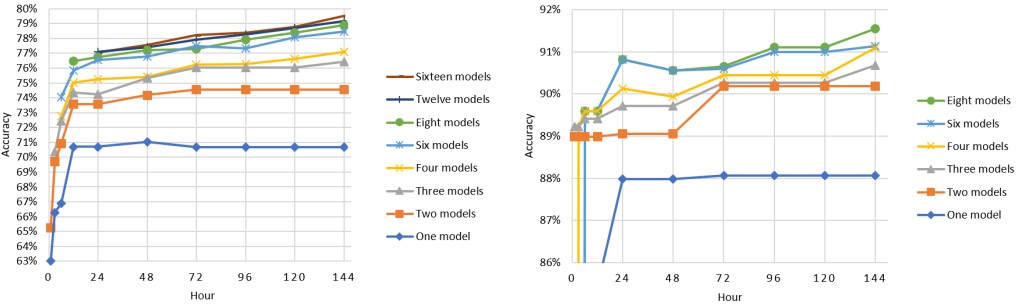

Figure 9: The CIFAR100 dataset          Figure 10: The microfossils dataset

Figure 11: Varying the max number of models in ensembles in function of Hyperband running time

## Conclusion

Due to the experimental nature of deep learning and the increasing of available computing power like multi-GPUs servers, it allows to sweep deep learning architectures. The standard usage of hyper parameter optimization consists in training hundreds of models and keeping only the best one, which leads to an important waste of energy.

We show that Hyperband efficiently generates diverse architectures coupled by a significant number of combinations between them. That is the reason why a smart selection of models with Dijkstra allows to build accurate ensembles. This workflow benefits of the increasing computational power and proposes a general approach to unify hyper-parameter search and resembling models. On two datasets, it has been also showed that our ensembles are more accurate than a naively build ensemble. Our workflow also yields an ensemble more accurate than any other ensemble of ResNet models.

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

# A ENSEMBLE AWARDS IN PAST PUBLIC IMAGE RECOGNITION CHALLENGES

URL : https://ndres.me/kaggle-past-solutions

- **Name: Understanding Clouds from Satellite Images**
- **Description:** Can you classify cloud structures from satellites?
- **Participation:** 1538 teams
- **Prize:** $10K
- **Dead line:** 2019-11-19
- **1st place** – Use averaging predictions of 3 segmentations models; averaging 9 segmentations models
- **3rd place** – Use majority vote of 4 segmentations models
- **4th place** – Averaging of 10 models
- **5th place** – Weighted averaging of 3 segmentations models

- **Name: RSNA Intracranial Hemorrhage Detection**
- **Description:** Identify acute intracranial hemorrhage and its subtypes
- **Participation:** 1345 teams
- **Prize:** $25K
- **Dead line:** 2019-10-28
- **2st place** – 15 bagging LSTMs (3 bootstraps)
- **3rd place** - Use weighted averaging predictions of 17 models
- **5th place** – Stacking of 9 segmentation models

- **Name: Lyft 3D Object Detection for Autonomous Vehicles**
- **Description:** Can you advance the state of the art in 3D object detection?
- **Participation:** 547 teams
- **Prize:** $25K
- **Dead line:** 2019-11-3
- **3rd place** – 3 faster-rcnn models with Soft NMS

- **Name: Severstal Steel Defect Detection**
- **Description:** Can you detect and classify defects in steel?
- **Participation:** 2431 teams
- **Prize:** $120K
- **Dead line:** 2019-10-18
- **1st place** – Ensemble of 4 classifications (which one ?); Ensemble of 9 segmentations(which one ?)
- **4th place** – Ensemble of 9 segmentations (which one ?)

- **Name: Kuzushiji Recognition**
- **Description:** Opening the door to a thousand years of Japanese culture
- **Participation:** 293 teams
- **Prize:** $15K
- **Dead line:** 2019-10-15
- **1st place** – Ensemble of 2 R-CNN
- **2nd place** – 1 Faster-RCNN, Stacking with an ensemble of XGBoost and LightGBM averaging
- **3rd place** – Hard voting of 5 models; NMS with 2 models

- **Name: The 3rd YouTube-8M Video Understanding Challenge**
- **Description:** Temporal localization of topics within video
- **Participation:** 283 teams
- **Prize:** $25K
- **Dead line:** 2019-10-04
- **1st place** – 17 averaging models + smooth out predictions
- **2nd place** – 7 models weighted averaging, weights fixed manually
- **3rd place** – Stacking of 12 models

- **Name: APTOS 2019 Blindness Detection**
- **Description:** Detect diabetic retinopathy to stop blindness before it's too late
- **Participation:** 2931 teams
- **Prize:** $50K
- **1st place** – Ensemble of 8 models with stacking
- **4th place** – Averaging of 3 models

# B    REPRODUCIBILITY

## B.1    HARDWARE

Computing nodes used are the same than the Oak Ridge Summit nodes. 6 Tesla-V100 GPUs was used for running each hyperparameter optimization algorithm and generate required populations.

## B.2    SOFTWARE STACK

Hyperparameter optimization framework Tune Liaw et al. (2018) was used. It runs above the Ray client-server Moritz et al. (2018) witch spread experiments to run on GPUs. Deep Learning training and data augmentation was coded with the framework Keras Gulli & Pal (2017) with backend Tensorflow 1.14.0 Abadi et al. (2015).

## B.3    THE TWO DATASET USED

**The CIFAR100 dataset.** CIFAR100 Giuste & Vizcarra (2020) consists to 60,000 32x32 RGB images in 100 classes. For each class, there are 580 training images, 20 validating images and 100 testing images.

**The Microfossils dataset.** Microfossils are extremely useful in age dating, correlation and paleo-environmental reconstruction to refine our knowledge of geology. Micro-fossil species are identified and counted on large microscope images and thanks to their frequencies we can compute the date of sedimentary rocks.

To do reliable statistics, a big amount of objects needs to be identified. That is why we need deep learning to automate this work. Today, between 400 and 800 fields of view (microscopy imagery) need to be shot for 1 rock sample. In each field of view, there are between 300 to 400 objects to identify. Among these objects, there are non-fossils (crystals, rock grains etc...) and others are fossils that we are looking for to study rocks.

Our dataset contains 91 classes of 224x224 RGB images (after homemade preprocessing). Micro-fossils are calcareous objects took with polarized light microscopy. The classes are unbalanced. We have from 50 images to 2500 images by class, with a total of 32K images in all the dataset. The F1 score was used and labeled as 'accuracy' on all benchmarks.

## B.4    HYPERPARAMETER CONFIGURATION SPACE

The table 5 shows all hyperparameters properties in this workflow. We use a ResNet Zagoruyko & Komodakis (2016) based architectures due to its simplicity to yield promising and robust models on many datasets . We explore different residual block versions: "V1", "V2" He et al. (2015) and "next" Xie et al. (2016). Regarding the optimization method, we use adam optimizer Kingma & Ba (2014) due to its well known performance and its low learning rate tuning requirement. Hyperparameters labeled as "mutable" can be updated during the training, for example the learning rate can change but the architecture cannot. PBT algorithm is the only one algorithm tested to discover a schedule of mutable hyperparameters.

We aware that our research may have a limitations. The range of hyper-parameter can be to short compared to good results found in the literature Zagoruyko & Komodakis (2016) like the batch size, width and depth of convolutionnal neural network. Moreover we could explore other optimization strategies like SGD with momentum and also the learning rate decay. Next, dropout is also a promising method we could explore. To finish, on CIFAR100 our maximum number of epochs is 100 and scientists before us usually use 160 epochs.

## B.5    ADAPTATION TO APPLY ON CIFAR100

The CIFAR100 dataset contains 32x32 images while usually ResNet are adapted to be used on imagenet (224x244 images). Those different resolution need some adaptation. On the CIFAR100, the first convolutionnal network is replaced from the 7x7 kernel size with a stride of 2, to a 3x3 kernel size with a stride of 1.

Table 5: The hyperparameter space experimented

|  | Name | Type | Range | Mutable ? |
|---|---|---|---|---|
| Optimization | Learning rate | Continuous | [0.001, 0.01] (log space) | yes |
|  | Batch size | Discrete | [8; 48] |  |
|  | L2 regularization factor | Continuous | [0;0.1] (log space) |  |
| Neural network architecture | Convolution type | Categorical | {"v1", "v2", "next"} | No |
|  | Activation function | Categorical | {"tanh", "relu", "elu"} |  |
|  | Number of filters in the first convolution layer | Discrete | 32, 64, 96, 128 |  |
|  | Number of filters in other convolution layers | Discrete | 32, 64, 96, 128 |  |
|  | Number of convolutionnal block in the first stage | Discrete | 1, 2, 3, 4, 6, 8, 11 |  |
|  | Number of convolutionnal block in the second stage | Discrete | 1, 2, 3, 4, 6, 8, 11 |  |
|  | Number of convolutionnal block in the third stage | Discrete | 1, 2, 3, 4, 6, 8, 11 |  |
|  | Number of convolutionnal block in the fourth stage | Discrete | 1, 2, 3, 4, 6, 8, 11 |  |
| Data augmentation | Max zoom | Continuous | [0; 0.6] | yes |
|  | Max translation | Continuous | [0; 0.6] |  |
|  | Max shearing | Continuous | [0; 0.3] |  |
|  | Max channel shifting | Continuous | [0; 0.3] |  |
|  | Max rotation measured in degrees | Discrete | [0; 90] (log space) |  |

