# OpenReview forum: "Recycling sub-optimial Hyperparameter Optimization models to generate efficient Ensemble Deep Learning"
_ICLR.cc/2021/Conference — Reject_

### Official Review · AnonReviewer4 · 2020-10-23
**Ensembles of deep models are created by first running Hyperband to sample HP configurations, and then run a greedy algorithm to find a good ensemble from the pool. Unfortunately, this is pretty well known already.**

**Rating:** 3
**Confidence:** 4

**Review:**

This paper is proposing to build ensembles of deep models, components of which have different hyperparameter (HP) configurations. This is done by first running Hyperband to create a large pool, and then run a greedy algorithm to construct an ensemble. This algorithm is termed Dykstra's algorithm on a certain graph, but it is of course simply just the default greedy algorithm, which is almost by default used to create an ensemble from a pool. The correct reference for this is [1], and this is just what people do when they create ensembles. The paper also misses a number of relevant recent work to build ensembles of deep models, at least [2, 3]. There is nothing new here, except maybe that Caruana's algorithm can now also be called Dykstra's.
In the very unlikely case I missed something, and what they call Dykstra's method here (not detailed in the paper) is different from the obvious greedy method of Caruana, then the paper fails by not comparing against this obvious baseline.

[1] Caruana, Niculescu-Mizil, Crew, Ksikes
  Ensemble selection from libraries of models
  ICML '04: Proceedings of the twenty-first international conference on Machine learning
[2] Deep ensembles:
  B. Lakshminarayanan, A. Pritzel, and C. Blundell. Simple and scalable predictive uncertainty estimation using deep ensembles. In Advances in Neural Information Processing Systems (NIPS), pages 6402–6413, 2017.
[3] Batch ensembles:
  Y. Wen, D. Tran, and J. Ba. Batch ensemble: an alternative approach to efficient ensemble and lifelong learning. arXiv preprint arXiv:2002.06715, 2020

---

### Official Review · AnonReviewer3 · 2020-10-27
**Paper not sufficiently well written and lacks of clear highlight on the contributions.**

**Rating:** 3
**Confidence:** 5

**Review:**

After rebuttal: no rebuttal, so I will keep my score.

-----------------------------------------------------------------------------------------------------------------------------------------------------------------------------

Overview: This paper proposes to apply ensemble methods to make use of sub-optimal configurations saved from Hyperband. In particular, the authors propose to use Dijkstra's algorithm to choose the best combination of size K among n models.

Reasons for the score:
Overall, I vote for reject. There are several reasons that will be detailed a bit in the next. Roughly, I think the paper requires a huge effort of rewriting before being published. Beyond that, I'm not fully convinced by the technical novelty or difficulty of this paper.

Pros:
- The idea of collecting sub-optimal configuration to boost the performance has not been considered in the context of HPO (to the best of my knowledge).
- The experiments in Table 1 and Table show that the workflow do improve the final performance.

Cons:
- My  first major concern is on the writing:
  - There is no clear problem formulation of HPO.
  - Personally, I'm not very comfortable with the authors not presenting formally what is Hyperband, what is Dijkstra's algorithm, and also how is the problem of finding the best ensemble modeled as a knapsack problem. I think these details can at least be provided in the appendices. I don't think it's appropriate to assume that readers are all familiar with every notion presented in the paper. Especially the main purpose of this work is to propose a new workflow that combines existing techniques, providing enough detail on how each component works and how they are connected in a precise way seems essential to me.
- Now regarding the experiments:
  - First of all, a general question, how many trials have you run for each experiment?
  - Also, I don't see error bars on figure 6, 7, 9, 10.
  - Could you explain a bit more why are you interested in the experiments of Table 2? I don't understand very well why it is useful to support the purpose of this work.
  - Could you please more precise on when do you stop the experiments? It looks like for some experiments, a fixed time horizon is given. I guess it's the same for others, but again, some precise and formal descriptions are more than welcomed.

General questions and remarks:
- Do you have any intuition on why you use averaging other than other ensemble strategies?
- Although ensemble has not been considered in the context of HPO, I'm not sure that the scientific contributions in this paper is significant enough. Could the authors highlight a bit more the technical difficulties if I have missed anything?
- In general, I feel like the paper is more like an engineering trick than a scientific discovery, which could be a nice contribution of course. But then in that case, I think more solid experiments should be provided. For example, in the context of HPO, we often want to see the evolution of performance over time, not only the final performance.

Minor comments and grammar issues (non-exhaustive):
- In general, I would suggest the authors to review a bit the writing style of the paper. Sometimes I feel like the authors have personal claims regarding some previous work without any support (which can be citations or even some intuitions). To cite one of them as example, in Section 3.2: Its pure-exploration nature combined with conservative resource allocation strategies can sweep better the hyperparameter space than other strategies like blackbox bayesian optimization. Maybe, but how is it compared to evolutionary algorithms for example? My point is that we should be careful about this kind of claims.
- The citation style is weird, authors can refer to Section 4.1 of the template file of ICLR.
- In the abstract: capable of doing sth. instead of capable to do sth.
- Section 2, paragraph Multi-objective goal, I don't really understand the sentence: Literature lets us imagine that the hyper-parameter function topology has two plateaus.
- Section 3.2: bayesian -> Bayesian.
- Section 3.3.3: hyperband -> Hyperband.
- Section 4.1: It shows very effective to detect... <- this is not correct grammatically.
- Section 4.3: Different combinations algorithms -> Different combinations of algorithms.
- Section 4.3: SP-MCTS do not falls into -> SP-MCTS does not fall into.

---

### Official Review · AnonReviewer2 · 2020-10-27
**Ensemble selection technique not compared to any existing one**

**Rating:** 4
**Confidence:** 4

**Review:**

The authors contribute the idea of selecting a subset of models for building ensembles using the Dijkstra algorithm. Furthermore they apply this algorithm on the models produced by Hyperband (HB) and also show that it performs well on other hyperparameter tuning methods, such as random search and Bayesian optimization.

Given that the latter is just an application of the algorithm, the former will have to be the main contribution of the paper.

What is unfortunately lacking is a comparison with other ensemble techniques including simpler ones. I am not an expert, but I know that simple greedy methods perform quite well, see for example [1]. So it it not clear the algorithm is any better than previous techniques. One advantage is that the proposed technique naturally incorporates a inference time constraint, but then it should be discussed.

Without a discussion and comparison to other widely using ensemble techniques this paper is incomplete and should not be accepted. The application alone on HPO technique is a fairly straightforward one, even if effective, and does alone not warrant a paper, with the decent but limited results presented. A stronger case could be made if for example the results allowed to reach top rank in many kaggle competitions, or in some existing AutoML competition.

#### Pros
* The paper is well written, polished, and easy to understand
* Combining hyperparameter tuning and ensembling is a good idea. Although already routinely done, for example in Kaggle competitions and in AutoML competitions, it deserves more attention and analysis.
* The results although limiting (only 2 datasets are used, ensemble baselines are missing) are promising.
* Incorporating the cost constraint is an important contribution, and should perhaps be discussed in more detail.


#### Cons
* There exist many ensemble selection techniques in the literature, that are simple and known to work well, for example greedily selecting the next method that performs best if combined with the currently selected ensembles. In some sense the proposed algorithm can be considered the obvious extension of this idea if presented with a cost constraint. But it would be good to better discuss how a cost constraint changes the situation.
* Too much focus on Hyperband, presented as part of the main contribution, I think the paper should focus more on their ensemble technique as it can be combined effectively with any method, as the results show, and just not that Hyperband seems to work best on the two datasets selected. Part of this seems to stem from Hyperband also working best when no ensembling is used, which is not necessarily in agreement with other HPO literature.
* The paper would be stronger if results were presented on more datasets.

[1] G Tsoumakas, I Partalas, I Vlahavas - A taxonomy and short review of ensemble selection

---

### Official Review · AnonReviewer1 · 2020-10-28
**No justification for the use of the Dijkstra algorithm**

**Rating:** 3
**Confidence:** 5

**Review:**


This paper proposes the use of Dijkstra's algorithm for ensemble construction, where the pool of classifiers to chose from is models trained during a hyperparameter optimization procedure. Authors show that the performance of the constructed ensembles improves upon the performance of a single classifier.

Strong points:

- Novel and interesting idea

Weak points:

- The paper would benefit significantly from proofreading and copyediting.
- The choice of the Dijkstra algorithm is poorly justified and might be incorrect
- More baselines for comparison are required

Recommendation:

I strongly recommend that this paper be rejected. While the idea of using Dijkstra for post hoc ensemble construction is interesting, I don't think it is justified and I am not convinced that it is a correct choice. Furthermore, the paper is riddled with mistakes, unneeded details and bad explanations. This paper needs to be entirely rewritten and the experiments require proper baselines for post hoc ensemble construction.

Extra comments:

There is no justification provided for using Dijkstra. The context is different from shortest path searching, as multiple classifiers in an ensemble have complex interactions that are lost via the majority voting and empirical risk estimation. In other words, the "weight" of the edge from one node to another will vary depending on the nodes visited beforehand, which is not a condition normally present for shortest path estimation. I think it means that it's incorrect to apply Dijkstra here, or at least that Dijkstra offers no guarantees. The same goes for the parallel with the Knapsack problem (i.e. I don't think this is an instance of the Knapsack problem).

Other baselines should be added, some simple ones would be forward/backward greedy search, stacking and genetic algorithms.

Section 3.3.1: showing with a single example that combining three models with a majority vote does not lead to optimal performance is hardly evidence that "smart algorithms are needed".

The related works section is needlessly long. For example, no need to mention work on multi-objective optimization.

Formatting of citations is incorrect, see the conference author guidelines.


Missing citations:

The correct citation to introduce boosting is "Y. Freund, and R. Schapire, “A Decision-Theoretic Generalization of On-Line Learning and an Application to Boosting”, 1997." not Schwenk & Bengio.

Amongst others, Caruana et al. (2004), Feurer et al. (2015) and Levesque et al. (2016) have already applied ensembling to hyperparameter optimization, such references are missing from your relevant works section.

References:

Caruana, R., Niculescu-Mizil, A., Crew, G., & Ksikes, A. (2004, July). Ensemble selection from libraries of models. In Proceedings of the twenty-first international conference on Machine learning (p. 18).

Feurer, M., Klein, A., Eggensperger, K., Springenberg, J., Blum, M., & Hutter, F. (2015). Efficient and robust automated machine learning. In Advances in neural information processing systems (pp. 2962-2970).

Lévesque, J. C., Gagné, C., & Sabourin, R. (2016). Bayesian hyperparameter optimization for ensemble learning. arXiv preprint arXiv:1605.06394.

---

### Decision · Program_Chairs · 2021-01-07
**Final Decision**

**Decision:**

Reject

**Comment:**

Following a strong consensus across the reviewers, the paper is recommended for rejection.
They have all acknowledged some weaknesses of the paper, for instance

* Inadequate reference to prior work
* Unsatisfactory level of polishing
* Too limited evaluation, with more comparisons to baselines required
* The proposed approach ("Dijkstra algorithm") is not enough justified and motivated
* Clarity (missing definitions of key components).

This list, together with the detailed comments of the reviewers, highlight opportunities to improve the manuscript for a future resubmission.